# RIP1 Is a Novel Component of γ-ionizing Radiation-Induced Invasion of Non-Small Cell Lung Cancer Cells

**DOI:** 10.3390/ijms21134584

**Published:** 2020-06-28

**Authors:** A-Ram Kang, Jeong Hyun Cho, Na-Gyeong Lee, Jie-Young Song, Sang-Gu Hwang, Dae-Hee Lee, Hong-Duck Um, Jong Kuk Park

**Affiliations:** 1Division of Radiation Biomedical Research, Korea Institute of Radiological and Medical Sciences, Seoul 01812, Korea; arbam0919@kirams.re.kr (A.-R.K.); jh_cho@kannwon.ac.kr (J.H.C.); ilr27387@kirams.re.kr (N.-G.L.); immu@kirams.re.kr (J.-Y.S.); sgh63@kirams.re.kr (S.-G.H.); hdum@kirams.re.kr (H.-D.U.); 2Department of Marine Food Science and Technology, Gangneung-Wonju National University, 120 Gangneung, Gangwon 210-702, Korea; neogene@gwnu.ac.kr

**Keywords:** signal transduction, γ-ionizing radiation, non-small cell lung cancer, cancer invasion, epithelial-mesenchymal transition, tumor microenvironment

## Abstract

Previously, we demonstrated that γ-ionizing radiation (IR) triggers the invasion/migration of A549 cells via activation of an EGFR–p38/ERK–STAT3/CREB-1–EMT pathway. Here, we have demonstrated the involvement of a novel intracellular signaling mechanism in γ-ionizing radiation (IR)-induced migration/invasion. Expression of receptor-interacting protein (RIP) 1 was initially increased upon exposure of A549, a non-small cell lung cancer (NSCLC) cell line, to IR. IR-induced RIP1 is located downstream of EGFR and involved in the expression/activity of matrix metalloproteases (MMP-2 and MMP-9) and vimentin, suggesting a role in epithelial-mesenchymal transition (EMT). Our experiments showed that IR-induced RIP1 sequentially induces Src-STAT3-EMT to promote invasion/migration. Inhibition of RIP1 kinase activity and expression blocked induction of EMT by IR and suppressed the levels and activities of MMP-2, MMP-9 and vimentin. IR-induced RIP1 activation was additionally associated with stimulation of the transcriptional factor NF-κB. Specifically, exposure to IR triggered NF-κB activation and inhibition of NF-κB suppressed IR-induced RIP1 expression, followed by a decrease in invasion/migration as well as EMT. Based on the collective results, we propose that IR concomitantly activates EGFR and NF-κB and subsequently triggers the RIP1–Src/STAT3–EMT pathway, ultimately promoting metastasis.

## 1. Introduction

Lung cancer is one of the most prevalent cancer types worldwide. Non-small cell lung cancer (NSCLC) represents the majority of cases, accounting for 85% of all lung cancers. In particular, the five-year survival rate for NSCLC is extremely low [1,2]. The current therapeutic strategies for NSCLC mainly involve radiotherapy, chemotherapy and surgery [3,4]. Radiotherapy is the principal treatment method applied to induce cell death for various cancer types. However, this option is associated with several side-effects, including decreased immunity owing to destruction of immune cells, resulting in reduced elimination by the immune system and subsequent resistance to therapy [5]. Therefore, besides having a therapeutic effect, radiotherapy can promote the malignant characteristics of cancer cells [6]. This IR-induced development of radiation resistance of cancer cells has contributed significantly to high mortality rates in patients. Members of the p53 and Bcl-2 families have been shown to regulate radioresistance of lung and pancreatic cancer by acting as modulators of cell death [7,8]. The main downstream target of epidermal growth factor receptor (EGFR) signaling, PI3K/Akt, plays a significant role in radioresistance of head-and-neck cancer [9]. However, the molecular mechanisms of action of these factors in NSCLC are yet to be elucidated.

Recent in vitro studies have suggested that in addition to its therapeutic effect, γ-ionizing radiation (IR) stimulates the invasiveness of different cancer types, such as glioma, hepatocellular carcinoma and pancreatic cancer, by inducing several intracellular signaling pathway cascades. In vivo studies have further confirmed that IR exposure at primary tumor sites promotes metastasis [10,11]. Advanced invasiveness is associated with increased activity and expression of MMP proteins stimulated by various intracellular survival signals, such as NF-κB, PI3k, AKT, JNK and MAPK. These proteins are survival factors that facilitate resistance to various stress conditions [12,13,14,15]. IR-induced MMP-9 is reported to be activated by PI3K/Akt, MAPK/ERK, JNK and p38 signaling [16]. A previous study by our group suggests that IR-induced epithelial-mesenchymal transition (EMT) is dependent on activation of EGFR–p38/ERK–STAT3/CREB-1 signaling in A549 cells [17]. Activities of these signaling molecules in A549 cells are enhanced upon exposure to IR, resulting in EMT progression, supporting their utility as novel biomarkers for indication of IR resistance. The efficiency of radiation treatment may therefore be effectively improved by identifying novel diagnostic biomarkers that endow resistance. In this study, we identified receptor interacting protein kinase (RIP, RIPK) 1 as a novel biomarker to determine radiation effect.

The RIP family is composed of 7 members, of which RIP1 is the first identified member [18]. RIP1 plays a critical role in cellular stress signaling that occurs in response to various intra-/intercellular factors, such as inflammation and DNA damage [19]. This protein activates several transcription factors that trigger the gene expression response to inflammatory stimuli and protect against cell death [20]. RIP1 is an important regulator of cell survival and a pivotal component of the inflammatory signaling pathway [21]. A recent study has demonstrated the involvement of RIP1 in invasion of gallbladder carcinoma [22]. NF-κB, a key participant in various signal transduction pathways, was also activated in cells exposed to IR in this study. As NF-κB and RIP1 are components of common inflammation-related pathways [23], we focused on the relationship between these two proteins in the mechanism underlying IR-induced invasion and EMT activation. We additionally investigated whether RIP1 plays a role in EGFR–p38/ERK–STAT3/CREB-1 signaling axis, which we reported previously [17], and the intervening location of RIP1 in the signaling pathway.

## 2. Results

### 2.1. IR Induces Invasion/Migration via Upregulation of EMT and RIP1

In a previous report, we showed that IR (10 Gy) treatment enhances the invasion/migration of A549 cells [17]. Data from the invasion and migration assays in the current study further established that IR (10 Gy) enhances invasion/migration of A549 cells by about 200%–300% (Figure 1A,B). However, we observed no differences in cell death and expression level of γ-H2AX, a representative marker for DNA-damage, between the IR-treated and control groups (Figure 1C).

To ascertain the potential mechanisms underlying IR-mediated invasion/migration, immunoblotting was performed to assess the expression levels of MMP-2, MMP-9 and vimentin. IR treatment clearly led to an increase in MMP-2, MMP-9 and vimentin levels (Figure 1D). Increased expressions of RIP1 and vimentin are detected even when NCI-H460 cells were irradiated with a dose of 2.5 Gy (Figure 1E).

As RIP1 is reported to be involved in invasion of gallbladder carcinoma cells [22], we further examined whether IR affects RIP1 expression. Immunofluorescence (IF) staining experiments revealed enhanced levels of RIP1 in A549 cells exposed to IR (Figure 1E), suggesting that the IR-induced invasion/migration of A549 cells is related to EMT induction and RIP1 expression. To evaluate the biological function of RIP1 in vivo, A549 cells were subcutaneously injected into nude mice and then exposed them to IR (10 Gy) for two days. Mice were sacrificed, and xenograft tissues were collected for immunohistochemical (IHC) and hematoxylin and eosin (H&E) staining. The IHC dataset shows that the expression of RIP1 is upregulated in IR-induced tumorigenisis in A549 cells (Figure 2A). In addition, we showed that vimentin was upregulated in IR-induced tissues compared to that in adjacent normal tissues (Figure 2A). Expression of RIP1 and vimentin proteins was measured by an IHC staining assay and was quantified. Results suggested that the IR-treated groups increased the score of RIP1 and vimentin (Figure 2B,C).

### 2.2. IR-Induced Invasion/Migration Is Mediated by the EGFR/Src/STAT3 Pathway

IR treatment (10 Gy) triggered activation of the EGFR pathway in A549 cells in our previous study [17]. In view of another recent report that TNF-related apoptosis-inducing ligand (TRAIL) activates the Src-STAT3 pathway to induce invasion/migration in NSCLC cells [24], we postulated the possibility of a relationship between EGFR and Src-STAT3 in the IR-induced increase in invasion/migration. To examine this hypothesis, immunoblotting was conducted to determine the expression levels of MMP-2, MMP-9, vimentin, p-EGFR, total EGFR, p-Src, total Src, p-STAT3 and total STAT3. Notably, IR treatment induced an increase in MMP-2, MMP-9, vimentin, p-EGFR, p-Src and p-STAT3 levels (Figure 3A). Next, we blocked EGFR through pre-treatment with a specific inhibitor, which led to a decrease in IR-induced RIP1 and an increase in p-Src and p-STAT3 levels (Figure 3B), suggesting that EGFR is located upstream of the Src-STAT3 pathway. Additionally, pre-treatment with Src and STAT3 inhibitors suppressed IR-induced phosphorylation of Src and STAT3, respectively, as well as expression of MMP-2, MMP-9 and vimentin (Figure 3C,D). These results support the theory that IR-induced invasion/migration is mediated via activation of an EGFR-Src-STAT3 pathway in vitro. We also performed xenograft experiments to prove the involvement of the EGFR-Src-STAT3 pathway in IR-induced invasion/migration in an in vivo system. As shown in Figure 3E–H, the collected xenograft tissues were analyzed for histology (H&E staining) and IHC for p-Src, Src, p-STAT3 and STAT3. The IHC dataset and score analyses of IHC show that the expression of p-STAT3 is upregulated in IR-irradiated xenografts made with A549 cells (Figure 3E,F). However, p-Src was barely detectable and there was no significant difference in the p-Src when 10 Gy-irradiated groups were compared with the control group (data not shown). These results indicate that Src phosphorylation might be not induced under in vivo conditions. Although Src phosphorylation was not evident in the in vivo study, the results of the in vitro study indicate that IR-induced invasion/migration might be mediated via activation of an EGFR-Src-STAT3 signaling axis in an in vivo system.

### 2.3. Inhibition of RIP1 Protein Suppresses Invasion/Migration in IR-Treated A549 Cells

Since RIP1 expression is stimulated by IR and related to the EMT pathway (Figure 1), we examined whether the kinase activity of the protein is involved in promotion of invasion/migration. Notably, treatment with the RIP1 kinase inhibitor, necrostatin (Nec), suppressed IR-induced invasion/migration (Figure 4A,B) but exerted no effects on cell viability in our system (Figure 4C). Nec inhibited EMT, and interestingly affected the EGFR-Src-STAT3 pathway (Figure 4D,E). While we observed no effect on RIP1 expression, treatment with Nec suppressed IR-induced phosphorylation of EGFR in addition to EMT progression (Figure 4D), supporting the involvement of RIP1 in EGFR phosphorylation. IF staining findings further confirmed that Nec treatment suppresses the IR-induced phosphorylation of Src and STAT3 (Figure 4E) but does not affect RIP1 expression (Figure 4F). Our results indicate that RIP1 kinase might play an important role in IR-induced invasion/migration, supporting the involvement of an EGFR/RIP1-Src-STAT3 pathway.

### 2.4. Knockdown of RIP1 Suppresses Invasion/Migration of IR-Treated A549 Cells

In view of the finding that inhibition of RIP1 kinase activity leads to a decrease in IR-induced invasion/migration, we further examined whether RIP1 knockdown with specific small interference RNA (siRNA) affects progression in IR-exposed A549 cells (Figure 5A,B). Knockdown of RIP1 did not affect cell viability (Figure 5C), indicative of no influence on cell survival, but suppressed EMT induction (Figure 5D). Interestingly, RIP1 depletion led to inhibition of IR-induced EGFR phosphorylation (Figure 5D), consistent with data obtained from RIP1 kinase inhibitor treatment experiments (Figure 4). Accordingly, we postulate that RIP1 and EGFR have a close functional and physical intracellular relationship within pathways activated upon IR exposure. IR-induced Src and STAT-3 phosphorylation were additionally suppressed upon siRNA-mediated knockdown of RIP1 (Figure 5D), consistent with the theory that RIP1 modulates the EGFR/RIP1-Src-STAT3 pathway. A decrease in IR-induced RIP1 expression following transfection of specific siRNA was detected using IF staining (Figure 5E). We propose that RIP1 expression, as well as kinase activity, is critically involved in IR-induced invasion/migration.

### 2.5. NF-κB Is Associated with RIP1 Expression and EMT in IR-Treated A549 Cells

NF-κB is located downstream of RIP1 in TNF-α signaling and known to integrate responses to various stimuli, including IR [25,26,27]. Accordingly, we examined whether NF-κB is additionally involved in IR-induced invasion/migration. Interestingly, IR exposure induced increased levels of p105, phosphorylated p65 and IκB (Figure 6D), as well as invasion/migration (Figure 6A,B) suggestive of NF-κB activation, coincident with previous reports [28]. Treatment with a specific pharmaceutical inhibitor of NF-κB, Bay 11-7082, suppressed IR-induced invasion/migration of A549 cells (Figure 6A,B) via blockage of the EMT pathway (Figure 6D) but did not affect cell viability (Figure 6C). Interestingly, the inhibitor also induced a decrease in IR-induced RIP1 protein expression, indicating effects of NF-κB on the stability or production of RIP1 protein (Figure 6D). Based on these collective findings, we suggest that a NF-κB-RIP1-EMT pathway is activated in cells exposed to IR. A decrease in RIP1 expression induced by the NF-κB inhibitor was also observed using IF staining (Figure 6E).

## 3. Discussion

A previous study by our group aimed to identify targets for improving the therapeutic efficiency of radiotherapy and overcoming IR-induced invasion/migration by focusing on modulating cancer-specific physiological changes, such as the tumor microenvironment (TME), following IR treatment [29], which resulted in the identification of a novel EGFR–p38/ERK–STAT3/CREB-1–EMT pathway. Considering previous findings, we attempted to identify other intracellular signaling pathways that may modulate the TME. The TME constitutes a major proportion of the cancer mass and is a heterogeneous tissue complex containing various cell types, such as immune cells, fibroblasts, endothelial cells, stromal cells and extracellular matrix (ECM) secreted by TME compartments [30,31]. Recently, several therapy-induced TME responses have received considerable attention, owing to their potential to enhance therapeutic efficiency or evoke resistance [32,33]. IR treatment has significant effects on the TME, including microvascular/endothelial cell damage, local ischemia and bystander effects due to diffusible damage signals (e.g., ROS), stromal cell modulation and immune cell recruitment [34]. The TME is associated with both intrinsic and acquired resistance to IR treatment. Intrinsic TME-mediated resistance is derived from a protective niche of cancer cells located within the bone marrow or central nervous system, whereas acquired resistance arises via immune responses, senescence, protective niche cells and modulation of the ECM (contributing to both primary and metastatic cancer) [33,34]. One of the main roles of the TME in cancer is to facilitate malignant progression and metastasis [35]. Many investigators have reported metastasis arising from primary cancers following irradiation [36,37]. Our group previously demonstrated that IR enhances invasion and metastasis of cancer cells in vitro and in vivo by activating specific intracellular signaling pathways and EMT [17,38]. Von Essen identified two types of IR-induced metastasis, specifically, that in which local irradiation of a primary cancer increases metastasis and that in which irradiated normal tissues promote specific localization of metastases [39]. Therefore, efforts to develop novel agents capable of radiosensitizing cancer cells should consider the need to both promote cancer cell death and block IR-induced metastasis. Among these combined therapies, the combination of immunotherapy with IR treatment could exhibit a potent therapeutic effect due to the synergistic effects of intensifying the anti-tumor immune response, resulting in extended survival of patients [40]. The current trends of research-led drug development are focusing on targeting various features of TME to control cancer progressions or modulations induced by IR. Several intracellular molecules involved in the immune response, hypoxia and fibrotic processes are considered the main sources of TME targets to improve radiotherapy efficiency. Here, we have identified a novel intracellular signaling molecule, RIP1, which plays an essential role in inflammation and other immune responses, as well as cell death [18,41]. Firstly, we detected an increase in RIP1 expression induced by IR, which appeared to require phosphorylation of EGFR, which contains mutations in exons 18–21 encoding the TK domain in lung cancer [42,43]. These mutations result in a heterozygous and amplified allele that activates the Ras/Raf/MEK and PI3K/AKT pathways, with critical roles in predicting the prognosis, pathogenesis, progression and oncogenic behavior of NSCLC [42,43].

Next, we established that a RIP1-Src-STAT3 pathway is also activated in response to IR. Previous experiments have shown that TRAIL treatment induces the RIP1-Src-STAT3 pathway, in turn, promoting invasion of NSCLC cells [24]. Our group additionally demonstrated that IR induces STAT3 activation while promoting Bcl-X_L_ accumulation-dependent invasion and EMT [11]. STAT3 can be activated by receptors with intrinsic tyrosine kinase activity (e.g., EGFR) and Src [44]. In the current study, specific inhibitors of EGFR and Src decreased IR-induced phosphorylation of STAT3, resulting in reduced invasion and suppression of EMT (Figure 2). The RIP1-Src-STAT3 pathway was activated following exposure of cells to IR, which promoted invasion via EMT induction. The common responses to TRAIL and IR indicate that this inflammatory signaling pathway plays an important role in TME and potentially presents a significant prognostic marker. NF-κB is known to integrate responses to various stimuli, including IR [25,26], and to stimulate growth, resistance, inflammation and survival of cancer cells [45,46,47,48]. Consistently, NF-κB induced RIP1 expression in IR-treated conditions and inhibition of NF-κB blocked the IR-induced increase in RIP1 expression and EMT in the present study (Figure 6), suggesting that NF-κB is located upstream of RIP1. Our findings collectively support the induction of a novel EGFR/NF-κB–RIP1–Src-STAT3–EMT pathway by IR, which stimulates the invasion/migration activities of NSCLC cells (Figure 7). RIP1 and its related intracellular signaling components may therefore present effective biomarkers for use in treatment of NSCLC.

## 4. Materials and Methods

### 4.1. Cell Culture and Chemical Reagents

The human NSCLC cell lines A549 and NCI-H460 were purchased from the American Type Culture Collection (Rockville, MD, USA) and incubated at 37 °C in a 5% CO_2_ incubator. Necrostatin (RIP1 inhibitor) and PP2 (Src inhibitor) were obtained from Sigma-Aldrich (St. Louis, MO, USA), C188-9 (STAT3 inhibitor III) from EMD Millipore Corp. (Billerica, MA, USA) and gefitinib (EGFR inhibitor) and BAY 11-7082 (NF-κB inhibitor) from Santa Cruz Biotechnology, Inc. (Dallas, TX, USA).

### 4.2. Treatment with γ-IR

For IR treatment of cells, A549 cells and NCI-H460 cells (5 × 10^5^) were seeded into 60 mm dishes and incubated overnight. IR exposure (10 Gy for A549 cells and 2.5 Gy for NCI-H460 cells) was performed using ^137^Cs as a radiation source (Atomic Energy of Canada, Ltd., Mississauga, ON, Canada). IR-treated cells were used for experiments after 24 h.

### 4.3. Propidium Iodide (PI) Uptake Assay

Cells were seeded on a 6-well plate at a density of 1 × 10^5^ cells/well and incubated with or without necrostatin (20 µM) and NF-κB inhibitor (10 nM) for 24 h. Treated cells were harvested by trypsinization, washed twice with cold phosphate-beffered saline (PBS), and resuspended in 500 μL of 5 μg/mL PI (Sigma-Aldrich). The apoptotic fraction was evaluated via flow cytometry on a FACSort instrument (Becton Dickinson, San Diego, CA, USA).

### 4.4. Migration and Invasion Assays

Migration and invasion assays were performed as described previously [17]. Collagen (3 μg/μL; Sigma-Aldrich, St. Louis, MO)-coated transwell plates (8 μm pore; Corning Inc., Corning, NY, USA) were used for detection of cell migration and matrigel-coated transwells (1 mg/mL; Invitrogen Life Technologies, Carlsbad, CA, USA) were used for detection of cell invasion. IR-treated or non-treated A549 cells (2 × 10^4^) in 200 μL serum-free medium were added to 0.1% bovine serum albumin and seeded onto the upper chamber. The lower chamber was filled with 1 mL of 20% fetal bovine serum (FBS) medium. After incubation for 24 h with 5% CO_2_ at 37 °C, cell staining was performed with Microscopy Hemacolor (Merck, Whitehouse Station, NJ, USA), according to the manufacturers’ protocols. Stained cells were counted under a microscope and statistical analyses were performed.

### 4.5. Immunoblot Analysis

Immunoblot analyses were conducted as described previously [17]. Membranes were probed with antibodies against MMP (matrix metalloproteinase)-2 and -9, phosphorylated EGFR, total EGFR, phosphorylated Src, total Src, phosphorylated STAT3, total STAT3, phosphorylated IκBα, phosphorylated p65, p105, γ-H2AX (Cell Signaling Technology, Inc., Beverly, MA, USA), vimentin and RIP1 (BD Pharmingen, San Diego, CA, USA) and β-actin as a loading control (Sigma-Aldrich, St. Louis, MO, USA).

### 4.6. Immunofluorescence (IF) Staining

Cells were seeded on 12-well culture plates that contained chamber slides and treated with IR or left untreated. Briefly, cells were washed with PBS and fixed with 4% paraformaldehyde for 20 min. After fixation, cells were lysed in 0.2% Triton X-100 for 10 min and incubated with an anti-RIP1 antibody (MBL, Nagoya, Japan) overnight, followed by a FITC-conjugated anti-mouse secondary antibody (Jackson Immuno Research, West Grove, PA, USA) for 2 h at room temperature. Nuclei were stained with mounting solution and images of stained cells were acquired under a LSM880 confocal microscope (Carl Zeiss, Germany).

### 4.7. Small Interference RNA and Transfection

The siRNA constructs for RIP1 and non-targeted control were obtained from Santa Cruz Biotechnology, Inc. (Dallas, TX, USA). A549 cells were transfected with 200 pmol of each siRNA using Lipofectamine 2000 (Invitrogen Life Technologies, Carlsbad, CA, USA) according to the manufacturer’s instructions. After 24 h of recovery, the transfectants were used for experiments.

### 4.8. Xenograft Tumorigenicity Assays and Immunohistochemistry

Logarithmically growing A549 cells were harvested and washed twice with PBS, and 1 × 10^7^ cells in 0.1 mL PBS were then injected subcutaneously into the right hinge leg of six-week-old BALB/cAnNCrj-nu/nu mice (Envigo, Cambridgeshire, UK). The experimental groups comprised 4 and 4 mice engrafted with the control and IR-(10Gy), respectively. Xenografts reaching more than 100 mm^3^ were treated with γ-IR at 10 Gy per day for 2 days. The mice were sacrificed and tumor tissues were collected for histological analysis. The slides were deparaffinized, rehydrated and heated for antigen retrieval before incubation with the antibodies. The slides were incubated with an anti-RIP1, anti-phospho-Src, anti-Src, anti-phospho-STAT3, anti-STAT3 and anti-vimentin antibody for 1 h at room temperature, treated with a broad-spectrum secondary antibody conjugated with HRP for 1 h, developed using DAB and mounted after counterstaining with hematoxylin. Sliced tissues were stained with hematoxylin & eosin solution and histological analysis was performed under a microscope. Tissue spots were analyzed by morphologic analysis and protein expression was evaluated using TMA lab (Image Scope Software; Aperio Technologies)

### 4.9. Statistical Analysis

Data were analyzed using GraphPad Prism software (La Jolla, CA, USA). The significance of differences between experimental groups was determined using Student’s *t*-test. Data were considered significant at *p*-values < 0.05. Individual *p*-values in figures are denoted by asterisks (*; *p* < 0.05, **; *p* < 0.01, ***; *p* < 0.001). The numbers above the points or bars in the graphs represent the means of three independent experiments and error bars signify standard deviation (SD).

## 5. Conclusions

IR induces RIP1 protein expression, and the RIP1 expression has no influence on the increase of cell death but increases cancer invasion via EMT induction. In the signaling study, EGFR and NF-κB, which are the upstream signaling components of RIP1, were found to be activated concomitantly by IR. The consequentially increased expression of RIP1 promotes the RIP1–Src/STAT3–EMT pathway, resulting in the promotion of radiation-induced cancer cell invasion.

## Figures and Tables

**Figure 1 ijms-21-04584-f001:**
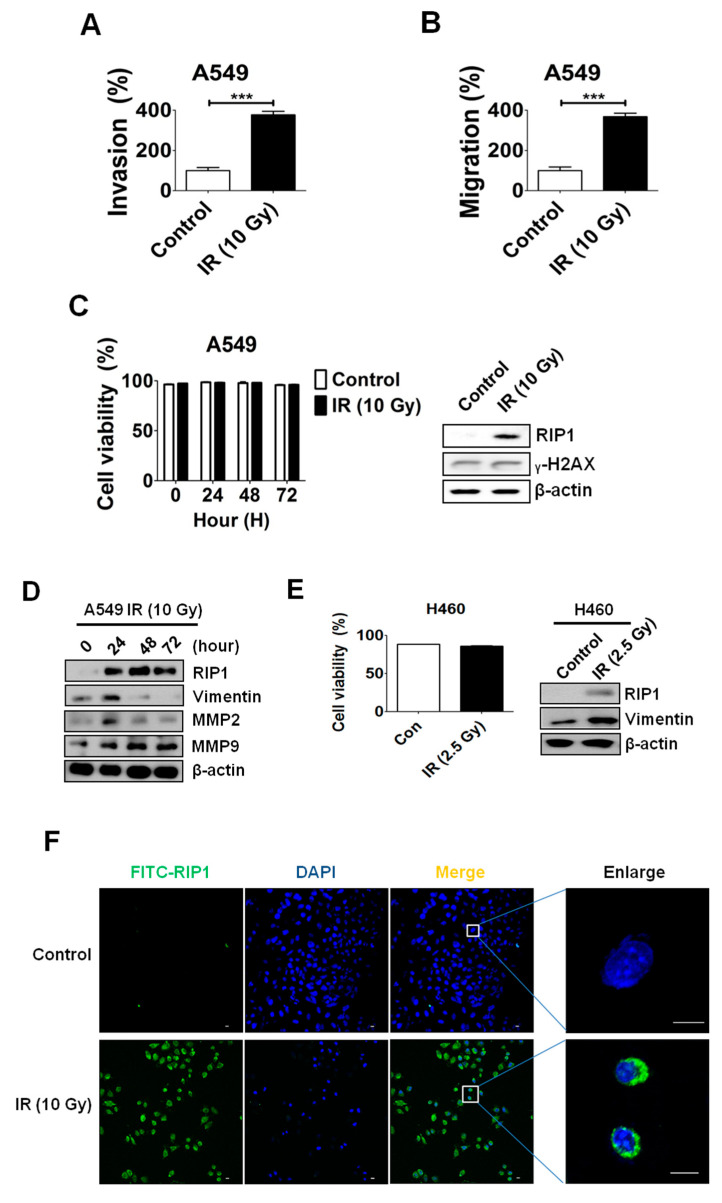
γ-ionizing radiation (IR) induces invasion/migration of A549 cells. (**A**) IR-induced invasion of A549 cells. A549 cells were seeded onto the upper chamber of a transwell system and cells on the membrane were counted after 24 h. (***; *p* < 0.001) (**B**) IR-induced migration of A549 cells. Experiments were performed as described in Materials and Methods. Representative data from triplicate experiments are shown on the right panel. (**C**) Cells were subjected to IR (10 Gy), and then apoptotic death and γ-H2AX expression was determined using the propidium iodide (PI) uptake assay and immunoblot analysis, respectively. Representative data from triplicate experiments are shown. (**D**) Lysates from A549 cells treated with IR (10 Gy) for the indicated periods were subjected to Western blot analysis for RIP1, vimentin, MMP-2, MMP-9 and β-actin (loading control). Data are representative of triplicate experiments. (**E**) cell viability and immunoblot analysis of NCI-H460 cells. (**F**) Immunofluorescence staining of RIP1-positive A549 cells after exposure to IR. Endogenous RIP1 expression in IR-treated or non-treated A549 cells was determined via immunofluorescence staining. A549 cells were stained with DAPI to visualize nuclei (blue) and immunolabeled with an anti-RIP1 antibody, which was detected via addition of FITC-conjugated IgG (green). Scale bar: 10 μm.

**Figure 2 ijms-21-04584-f002:**
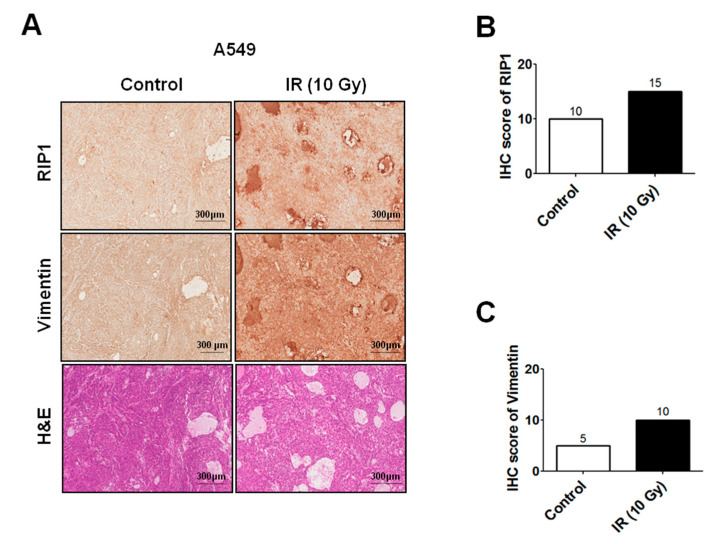
IR induces RIP1 and vimentin expression in a xenograft mouse model. (**A**) A549-tumor xenografts in response to sham irradiation or irradiation with 10 Gy. Tumors were harvested 48 h after irradiation. Immunohistochemical (IHC) analysis of xenografts tissues of mice after irradiation was performed with antibodies against RIP1 and vimentin. Hematoxylin and eosin (H&E) staining of xenografts tissues in mice after irradiation with 10 Gy. Scale bar indicates 300 μm. (**B**,**C**). The IHC score for RIP1, vimentin is shown in (**B**,**C**), respectively. The correlation plot of IHC-score quantification for RIP1 and vimentin.

**Figure 3 ijms-21-04584-f003:**
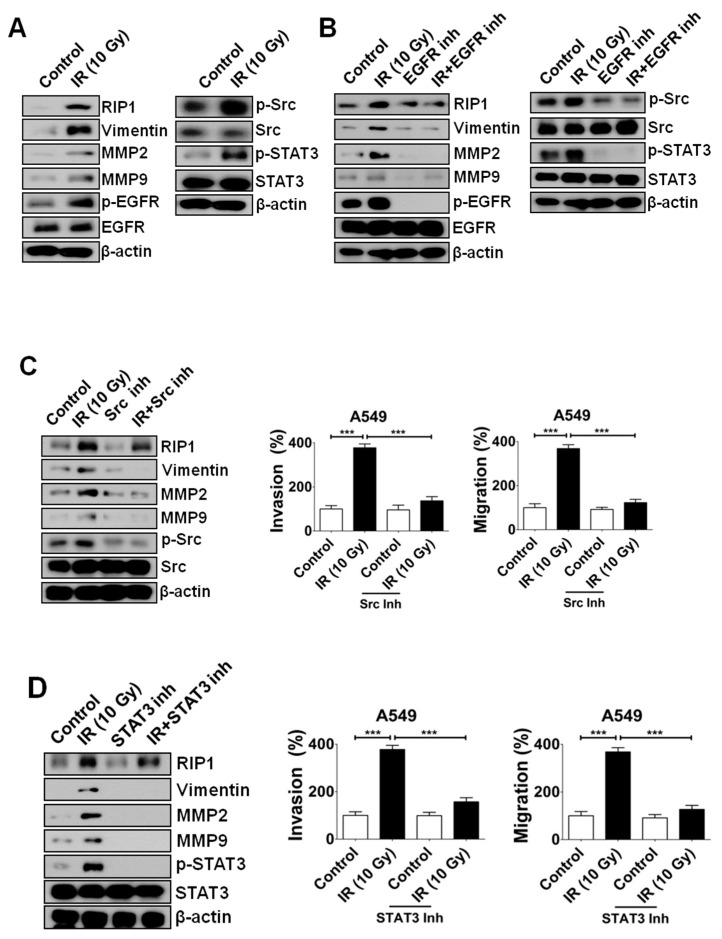
IR induces RIP1 expression and activates epithelial-mesenchymal transition (EMT) in vitro/in vivo. (**A**) Lysates from A549 cells treated with IR (10 Gy) were subjected to immunoblotting to examine the expression levels of RIP1, MMP-2, MMP-9, EGFR, p-EGFR, Src, p-Src, STAT3 p-STAT3, vimentin and β-actin (loading control). Representative data from triplicate experiments are shown. (**B**) A549 cells were exposed to IR (10 Gy) and treated with gefitinib (EGFR inhibitor, 10 µM). Immunoblotting was applied to examine the expression of RIP1, MMP-2, MMP-9, EGFR, p-EGFR, Src, p-Src, STAT3, p-STAT3, vimentin and β-actin (loading control). Representative data from triplicate experiments are shown. (**C**,**D**) A549 cells subjected to IR (10 Gy) were treated with Src (20 µM) or STAT3 inhibitor (20 µM) for the indicated periods and Western blot analyses for RIP1, vimentin, MMP-2, MMP-9, Src, p-Src, STAT3, p-STAT3 and β-actin (loading control) were conducted. Data are representative of triplicate experiments. (***; *p* < 0.001) (**E**) A549-tumor xenografts in response to sham irradiation or irradiation with 10 Gy. Tumors were harvested 48 h after irradiation. Immunohistochemical (IHC) analysis of xenograft tissues of mice after irradiation was performed with antibodies against p-STAT3, STAT3 and Src. Hematoxylin and eosin (H&E) staining of xenograft tissues in mice after irradiation with 10Gy. Scale bar indicates 300 μm. (**F**–**H**) The correlation plot of IHC-score quantification for p-STAT3, STAT3 and Src. The IHC scores for p-STAT3, STAT3 are shown in (**F**,**G**), respectively. The IHC score for Src is shown in (**H**).

**Figure 4 ijms-21-04584-f004:**
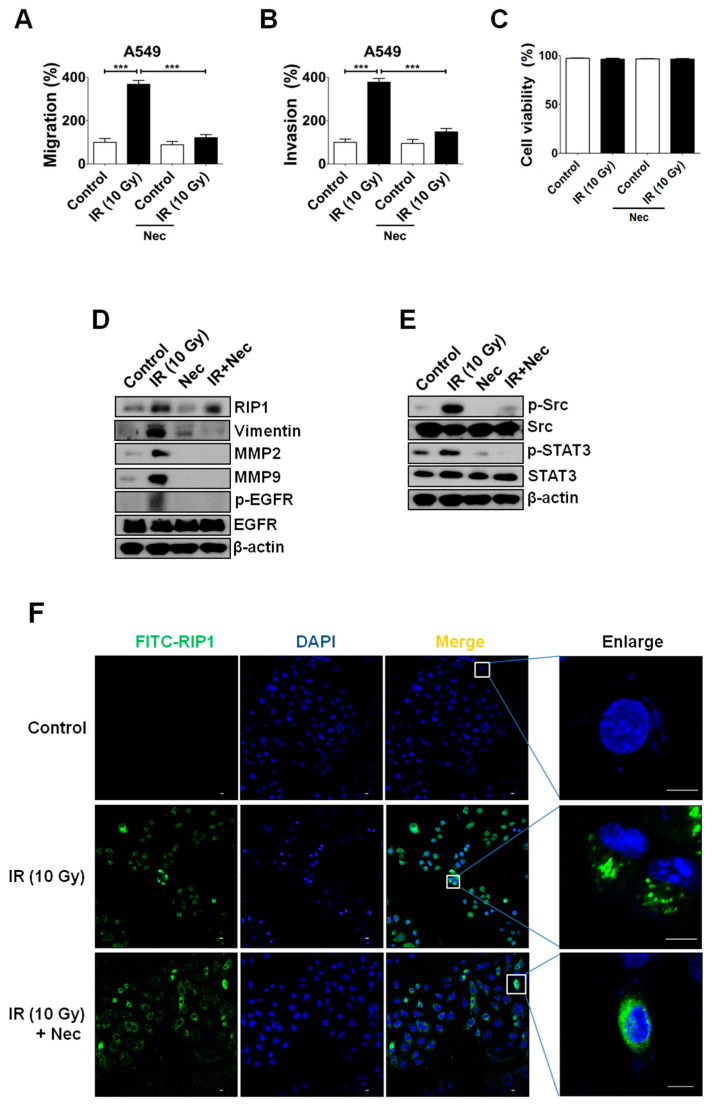
IR-mediated activation of EMT involves RIP1 kinase activity. (**A**) To determine the mechanism underlying IR-induced invasion of A549 cells, necrostatin was added to the lower chamber of a transwell system and cells were seeded onto the upper chamber. After 24 h, cells on the membrane were counted. (***; *p* < 0.001) (**B**) IR-induced migration of A549 cells. Experiments were performed as described in Materials and Methods. Representative data from triplicate experiments are shown on the right panel. (***; *p* < 0.001) (**C**) Cells were subjected to IR (10 Gy) and treated with necrostatin (20 µm), and apoptotic cell death determined via the PI uptake assay. Representative data from triplicate experiments are shown. (**D**,**E**) Immunoblotting for RIP1, MMP-2, MMP-9 EGFR, p-EGFR, Src, p-Src, STAT3 p-STAT3, vimentin and β-actin (loading control). (**F**) Immunofluorescence staining of RIP1-positive A549 cells after treatment with IR. Endogenous RIP1 expression in IR-treated or non-treated A549 cells was determined via immunofluorescence staining. A549 cells were stained with DAPI to visualize nuclei (blue) and immunolabeled with an anti-RIP1 antibody, which was detected with FITC-conjugated IgG (green). Scale bar: 10 μm.

**Figure 5 ijms-21-04584-f005:**
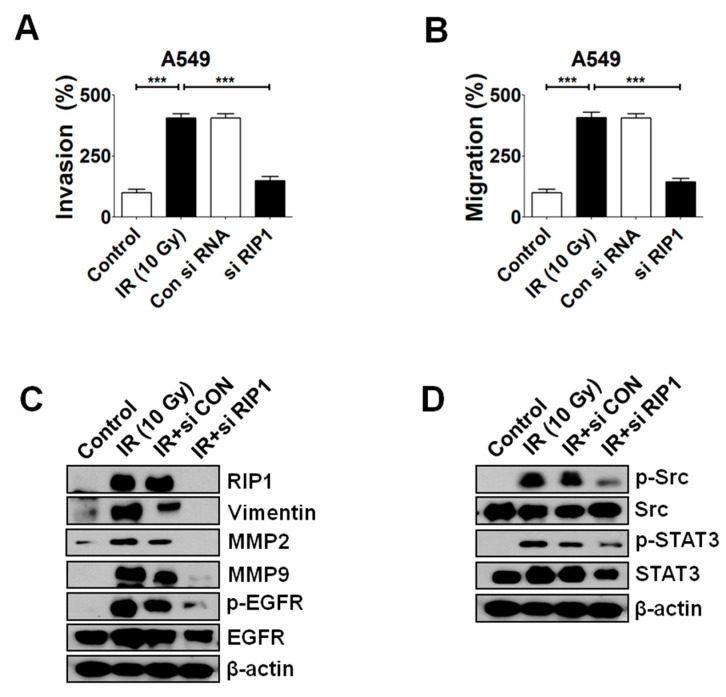
IR induces invasion/migration by promoting EMT via RIP1 activation. (**A**,**B**) A549 cells transfected with control or RIP1 siRNA for 24 h were subjected to invasion and migration assays. (***; *p* < 0.001) (**C**,**D**) Lysates from A549 cells transfected with control or RIP1 siRNA for 24 h were subjected to Western blot analysis for RIP1, MMP-2, MMP-9 EGFR, p-EGFR, Src, p-Src, STAT3 p-STAT3, vimentin and β-actin (loading control). (**E**) Immunofluorescence staining of RIP1-positive A549 cells after IR exposure. Endogenous RIP1 expression in IR treated or non-treated A549 cells was determined via immunofluorescence staining. A549 cells were stained with DAPI to visualize nuclei (blue) and immunolabeled with an anti-RIP1 antibody, which was detected with FITC-conjugated IgG (green). Scale bar: 10 μm.

**Figure 6 ijms-21-04584-f006:**
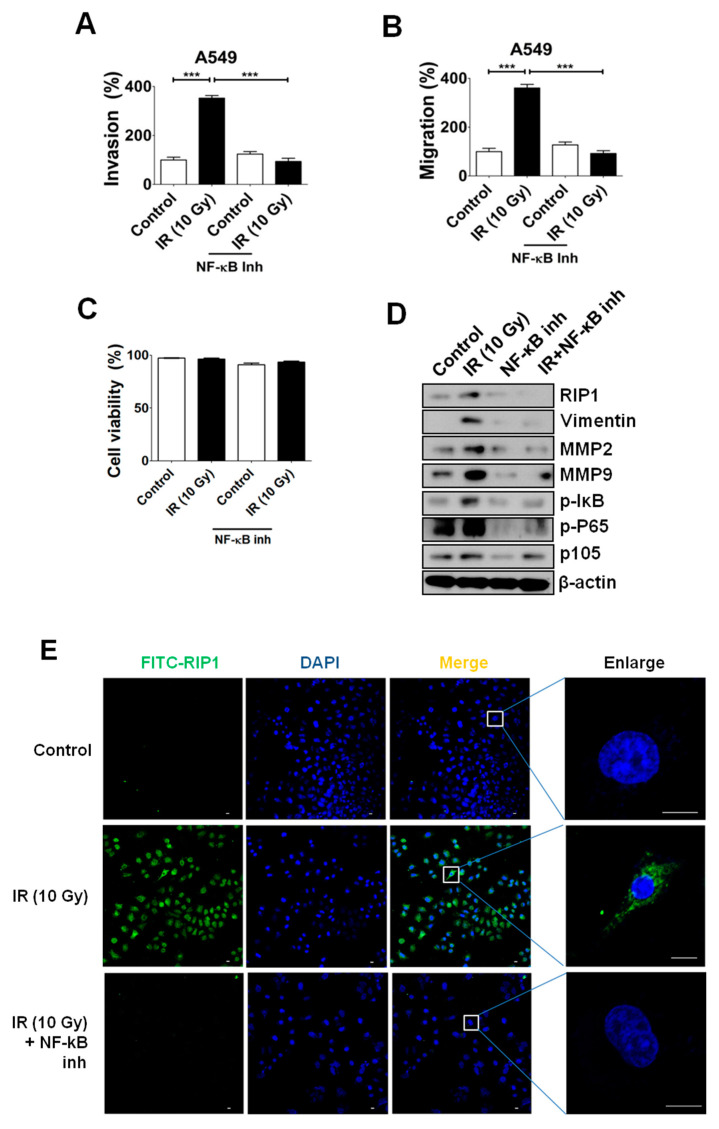
Expression of RIP1 and EMT progression is induced by NF-κB following IR exposure. (**A**) To examine the mechanisms underlying IR-induced invasion of A549 cells, necrostatin was added to the lower chamber of a transwell system and cells were seeded onto the upper chamber. After 24 h, cells on the membrane were counted. (***; *p* < 0.001) (**B**) IR-induced migration of A549 cells. Experiments were performed as described in Materials and Methods. Representative data from triplicate experiments are shown on the right panel. (***; *p* < 0.001) (**C**) Cells were subjected to IR (10 Gy) and treated with BAY 11-7082 (NF-κB inhibitor), and apoptotic death was determined using the PI uptake assay. Representative data from triplicate experiments are shown. (**D**) A549 cells were subjected to IR (10 Gy) and treated with BAY 11-7082 (NF-κB inhibitor, 10 nM). Expression levels of RIP1, MMP-2, MMP-9, IκB-α p-P65, p105, vimentin and β-actin (loading control) were determined via immunoblotting. Representative data from triplicate experiments are shown. (**E**) Immunofluorescence staining of RIP1-positive A549 cells following exposure to IR. Endogenous RIP1 expression in IR-treated and non-treated A549 cells was determined. A549 cells were stained with DAPI to visualize nuclei (blue) and immunolabeled with an anti-RIP1 antibody, which was detected with FITC-conjugated IgG (green). Scale bar: 10 μm.

**Figure 7 ijms-21-04584-f007:**
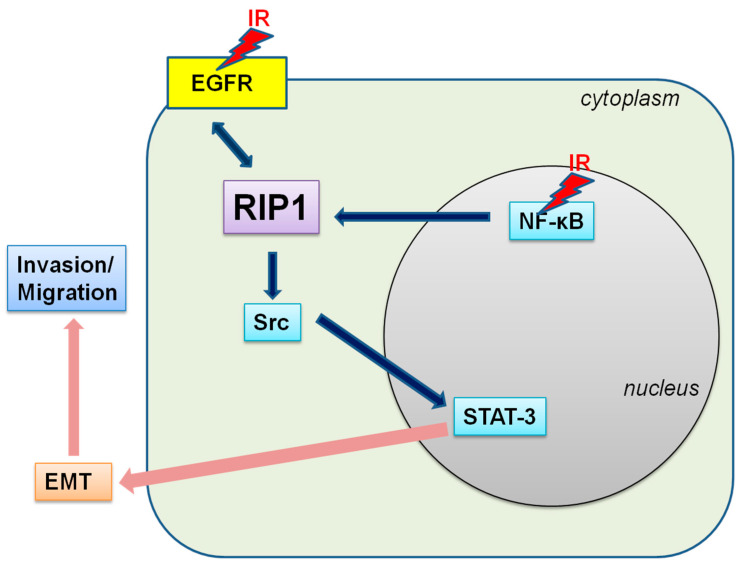
Scheme of the potential IR-induced EGFR/NF-κB–RIP1–Src–STAT3–EMT pathway.

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
