# Peer review of "RIP1 Is a Novel Component of γ-ionizing Radiation-Induced Invasion of Non-Small Cell Lung Cancer Cells"

_ijms, 2020, doi:10.3390/ijms21134584_

Round 1

Reviewer 1 Report

Authors have responded to some of the points this reviewer raised, but they don’t agree to examine the RIP1’s relationship with p38 and cancer stemness issues.  Both issues are closely related with authors previous works and even if authors would like to focus on the first and new findings, the both issues are important for the characterization of the new findings. This reviewer recommends to proceed these points.

Author Response

  • Authors have responded to some of the points this reviewer raised, but they don’t agree to examine the RIP1’s relationship with p38 and cancer stemness issues. Both issues are closely related with authors previous works and even if authors would like to focus on the first and new findings, the both issues are important for the characterization of the new findings. This reviewer recommends to proceed these points.

    -> We agree that there could be a correlation between p38 and cancer stemness. Actually, we are focused on this issue in our other novel project to reveal relationship between RIP1, MAPKinase activation, and cancer stemness. Thank you for your scientific advice.

Reviewer 2 Report

The authors have addressed all my questions. 

The authors should double check the grammar and then manuscript should be good to go.

Author Response

The authors have addressed all my questions.

The authors should double check the grammar and then manuscript should be good to go.

 -> Thank you for your positive evaluation. We will consider double checking of grammar.

Reviewer 3 Report

Authors have revised the manuscript satisfactorily. No further comments .

Author Response

Authors have revised the manuscript satisfactorily. No further comments

 -> Thank you for your positive evaluation.

This manuscript is a resubmission of an earlier submission. The following is a list of the peer review reports and author responses from that submission.

Round 1

Reviewer 1 Report

This paper is showing the importance of RIP1 in the IR induced invasiveness of lung cancer cells. Essentially this is interesting paper, but there are some concerns listed below.

Authors need to confirm and show that some epithelial and mesenchymal markers are changing with IR, if they would like to show the novel axis involving EMT.  Also authors need to show the markers are regulated by RIP1 activity, using over-expression, KD condition and inhibitors.

Some figures, especially Fig. 4, 5, are using oversize font and not well organized.  Authors have to be careful of these quality.

The immunostaining figures in Fig. 5 and Fig. 6 are not in good quality, and need to be improved.

Authors need to show the mechanical relationship between p38 and RIP1 in their IR induced system.  

IR might affect on stemless status of cancer, which is much related with EMT process. In the system authors are using, it should be important to know the stemless status at each condition.

Author Response

This paper is showing the importance of RIP1 in the IR induced invasiveness of lung cancer cells. Essentially this is interesting paper, but there are some concerns listed below.

Authors need to confirm and show that some epithelial and mesenchymal markers are changing with IR, if they would like to show the novel axis involving EMT. 

è We agree with this comment. Accordingly, we examined the expression levels of MMP proteins and vimentin, a representative EMT marker, to identify the status of the cells (Figure 1).

Also authors need to show the markers are regulated by RIP1 activity, using over-expression, KD condition and inhibitors.

è To address the reviewer’s comment, we performed RIP1 activity experiments (Figure 4) using a RIP1 inhibitor (necrostatin-1), and RIP1 expression modulation experiments (Figure 5) using RIP siRNA treatment.

Some figures, especially Fig. 4, 5, are using oversize font and not well organized.  Authors have to be careful of these quality.

è We have corrected the fonts and organization in Figures 4 and 5.

The immunostaining figures in Fig. 5 and Fig. 6 are not in good quality, and need to be improved.

è We repeated the immunostaining experiments and replaced the images in Figures 5 and 6.

Authors need to show the mechanical relationship between p38 and RIP1 in their IR induced system.  

è We agree that this is a relevant relationship. However, we previously studied IR-induced p38 and ERK activation, as mentioned in the original version of this manuscript (reference 17: Cho J.H. et al. Γ-Ionizing radiation induced activation of the EGFR/p38/ERK/STAT3/CREB1EMT pathway promotes the migration/invasion of non small cell lung cancer cells and is inhibited by podophyllotoxin acetate. Tumor Biol 2016). We did not mention MAPKs such as p38, ERK and JNK in the present manuscript because we wanted to focus on the relationship between RIP1 and Src/STAT3.

IR might affect on stemless status of cancer, which is much related with EMT process. In the system authors are using, it should be important to know the stemless status at each condition.

è We agree that this is an important concept. However, in this study, we wanted to focus on finding the novel links that we couldn’t identify in our previous study (reference 17), and found that RIP1 could be one of the components of the missing links.

Reviewer 2 Report

In this study, the authors demonstrated the upregulation of RIP1 in A549 human NSCLC cells in response to gamma-ionization radiation, and subsequent induction of

 Src-STAT3-EMT that promote cell invasion and metastasis.

Major points:

What was the clinical relevant of choosing gamma-IR at the dose of 10 Gy? What would the authors anticipate if the dose of the ganna-IR that would result in 10% ~ 20% cell killing were used?

Line 321 – 322 on Page 12. “The experimental groups comprised 2 and 4 mice engrafted with the control control and IR-(10Gy), respectively.” It was unclear what the authors were trying to say here. Were there 2 mice in the control group and 4 in the gamma-IR group. If so, the number of animals in the control group was not enough to provide statistically meaningful data.

Section 4.8: The tumor xenograft samples obtained from the in vivo study should be subjected to the Western blot analysis in order to verify the proposed mechanism based on the in vitro finding. In other words, determination of protein expression of p-Src, Src, p-STAT3 and STAT3 in tumor xenografts should be included.  

Minor points. Figures 1, 4 and 5 are not displayed properly.  

Author Response

Comments and Suggestions for Authors

In this study, the authors demonstrated the upregulation of RIP1 in A549 human NSCLC cells in response to gamma-ionization radiation, and subsequent induction of

 Src-STAT3-EMT that promote cell invasion and metastasis.

Major points:

What was the clinical relevant of choosing gamma-IR at the dose of 10 Gy? What would the authors anticipate if the dose of the ganna-IR that would result in 10% ~ 20% cell killing were used?

è We used this experimental condition in our previous report (reference 17: Cho J.H. et al. Γ-Ionizing radiation induced activation of the EGFR/p38/ERK/STAT3/CREB1EMT pathway promotes the migration/invasion of non-small cell lung cancer cells and is inhibited by podophyllotoxin acetate. Tumor Biol 2016).

Line 321 – 322 on Page 12. “The experimental groups comprised 2 and 4 mice engrafted with the control control and IR-(10Gy), respectively.” It was unclear what the authors were trying to say here. Were there 2 mice in the control group and 4 in the gamma-IR group. If so, the number of animals in the control group was not enough to provide statistically meaningful data.

è We repeated the animal experiments with four mice in each group, and corrected the manuscript accordingly.

Section 4.8: The tumor xenograft samples obtained from the in vivo study should be subjected to the Western blot analysis in order to verify the proposed mechanism based on the in vitro finding. In other words, determination of protein expression of p-Src, Src, p-STAT3 and STAT3 in tumor xenografts should be included.

è We additionally observed the protein expression levels of p-Src, Src, p-STAT3 and STAT3. This work is now presented in the Results section and Figure 3E-H.

Minor points. Figures 1, 4 and 5 are not displayed properly.

We repeated the immunostaining experiments and replaced the images in Figures 1, 4 and 5.

Reviewer 3 Report

In this manuscript authors have highlighted the relationship between NF-kB RIP1 in IR induced migration/invasion of lung cancer cells. Authors demonstrated the role of NF-kB/RIP1 axis via cell line model (A549) and animal xenografts of A549.

The data presentation is sound and it supports most of the arguments made by the authors. However, there are several questions which remain to be answered. 

Novelty: The introduction section reports that majority of the findings in this manuscript are either reported in lung cancer (by another group) or in another cancer. Therefore, it is critical that the authors provide a justification that what is novel in their findings? What was the knowledge gap that the findings in this manuscript are supposed to fill? In the last section of introduction authors should include a section focused on things that were not known and how the data in this manuscript would answer those questions. Currently, the manuscript lacks the novelty aspect.  Data/figure labels: They were all out order hence it the data presented in the figures (especially WB) could not be reviewed to it's best. Supplementary: authors state in the M&M that the full blot scans are available in supplementary file however, the supplementary file was not available in the system. Was it not uploaded by the authors? or was it an error on the editorial side? Role of immune system: authors focus on the role of immune system in IR resistance/migration however, the animal xenograft data was obtained from an immunocompromised animal model. Then, how the data is even relevant? if the authors are indicating a big role of immune system in IR resistance and migration then they must provide data supporting their view in an appropriate model system. Therefore, the animal data/in-vitro data is not supportive for the arguments involving immune system. An immunocompromised animal already mimics an irradiated situation to some extent (compromised immune system) hence this system would be very similar to an in-vitro assay. What additional information is obtained here? In addition, authors didn't show any migration/invasion phenotype in animal experiments (which again would depend on an appropriate model). Missing controls: IR would induce DNA damage therefore authors should show controls, IF of DNA damage response elements (such as gamma H2AX). Only one cell line: The whole manuscript is based on the data from one cell line. Currently, data obtained form a single cell line model is under great scrutiny. Authors should repeat some critical (if not all) experiments in another NSCLC cell line to support their data. 

Author Response

Comments and Suggestions for Authors

In this manuscript authors have highlighted the relationship between NF-kB RIP1 in IR induced migration/invasion of lung cancer cells. Authors demonstrated the role of NF-kB/RIP1 axis via cell line model (A549) and animal xenografts of A549.

The data presentation is sound and it supports most of the arguments made by the authors. However, there are several questions which remain to be answered. 

Novelty: The introduction section reports that majority of the findings in this manuscript are either reported in lung cancer (by another group) or in another cancer. Therefore, it is critical that the authors provide a justification that what is novel in their findings? What was the knowledge gap that the findings in this manuscript are supposed to fill? In the last section of introduction authors should include a section focused on things that were not known and how the data in this manuscript would answer those questions. Currently, the manuscript lacks the novelty aspect. 

è We agree with the reviewer’s points. We previously reported that IR activates the EGFR/p38/ERK/STAT3/CREB1EMT pathway and promotes the migration/invasion of non-small cell lung cancer cells (reference 17). In the present work, we focused on finding the novel links in the EGFR-mediated EMT induced by irradiation which was not identified in our previous report.  With this study, RIP1 was newly identified as a linker in the signaling pathway. We thus focused on showing the evidence of this linkage and performing functional analyses of RIP1 in our system. To address the reviewer’s point, we replaced the final sentences of the Introduction with new material that clarifies the novelty of the present study.

Data/figure labels: They were all out order hence it the data presented in the figures (especially WB) could not be reviewed to it's best.

è We apologize for this confusion, and have revised the fonts, sizes and WB locations in Figures 3A, 3B, 4C and 4D.

Supplementary: authors state in the M&M that the full blot scans are available in supplementary file however, the supplementary file was not available in the system. Was it not uploaded by the authors? or was it an error on the editorial side?

è We apologize for this error. This sentence has been deleted.

Role of immune system: authors focus on the role of immune system in IR resistance/migration however, the animal xenograft data was obtained from an immunocompromised animal model. Then, how the data is even relevant? if the authors are indicating a big role of immune system in IR resistance and migration then they must provide data supporting their view in an appropriate model system. Therefore, the animal data/in-vitro data is not supportive for the arguments involving immune system. An immunocompromised animal already mimics an irradiated situation to some extent (compromised immune system) hence this system would be very similar to an in-vitro assay. What additional information is obtained here?

è We agree with this comment. Accordingly, we deleted the immune system-related material from the Discussion and omitted original references 40 and 41.

In addition, authors didn't show any migration/invasion phenotype in animal experiments (which again would depend on an appropriate model). Missing controls: IR would induce DNA damage therefore authors should show controls, IF of DNA damage response elements (such as gamma H2AX). Only one cell line: The whole manuscript is based on the data from one cell line. Currently, data obtained form a single cell line model is under great scrutiny. Authors should repeat some critical (if not all) experiments in another NSCLC cell line to support their data. 

è To address the reviewer’s point, we now show expressional data for gamma H2AX in Figure 1C, right panel.

è We added viability and immunoblot analyses showing that the expression levels of RIP1 and vimentin expression are increase by IR in an additional NSCLC cell line (NCI-H460 cells). These results are presented in Figure 1E.